# New Chemicals Suppressing SARS-CoV-2 Replication in Cell Culture

**DOI:** 10.3390/molecules27175732

**Published:** 2022-09-05

**Authors:** Alexey Sulimov, Ivan Ilin, Danil Kutov, Khidmet Shikhaliev, Dmitriy Shcherbakov, Oleg Pyankov, Nadezhda Stolpovskaya, Svetlana Medvedeva, Vladimir Sulimov

**Affiliations:** 1Dimonta Ltd., 15 Nagornaya Str., Bldg 8, 117186 Moscow, Russia; 2Research Computing Center, Lomonosov Moscow State University, Leninskie Gory, 1, Building 4, 119234 Moscow, Russia; 3Department of Organic Chemistry, Faculty of Chemistry, Voronezh State University, 1 Universitetskaya Sq., 394018 Voronezh, Russia; 4State Research Centre of Virology and Biotechnology “Vector”, 630559 Koltsovo, Russia

**Keywords:** docking, quantum chemistry, SARS-CoV-2 replication, main protease, inhibitors, cell culture, 62P10, 81V55, 92C05, 92C40, 92C50, 65K10, 92C45

## Abstract

Candidates to being inhibitors of the main protease (M${}^{pro}$) of SARS-CoV-2 were selected from the database of Voronezh State University using molecular modeling. The database contained approximately 19,000 compounds represented by more than 41,000 ligand conformers. These ligands were docked into M${}^{pro}$ using the SOL docking program. For one thousand ligands with best values of the SOL score, the protein–ligand binding enthalpy was calculated by the PM7 quantum-chemical method with the COSMO solvent model. Using the SOL score and the calculated protein–ligand binding enthalpies, eighteen compounds were selected for the experiments. Several of these inhibitors suppressed the replication of the coronavirus in cell culture, and we used the best three among them in the search for chemical analogs. Selection among analogs using the same procedure followed by experiments led to identification of seven inhibitors of the SARS-CoV-2 replication in cell culture with EC${}_{50}$ values at the micromolar level. The identified inhibitors belong to three chemical classes. The three inhibitors, 4,4-dimethyldithioquinoline derivatives, inhibit SARS-CoV-2 replication in Vero E6 cell culture just as effectively as the best published non-covalent inhibitors, and show low cytotoxicity. These results open up a possibility to develop antiviral drugs against the SARS-CoV-2 coronavirus.

## 1. Introduction

Since the beginning of the COVID-19 pandemic, there have been many publications devoted to the computer search for inhibitors of SARS-CoV-2 therapeutic target proteins [1]. Among several therapeutic targets [2,3,4], the main protease (M${}^{pro}$) of SARS-CoV-2 is the most popular target. There are several high-resolution and good-quality 3D structures of M${}^{pro}$ in Protein Data Bank crystallized together with different ligands, a technology of expression and purification of the recombinant viral M${}^{pro}$ is well established and the fluorescence resonance energy transfer (FRET)-based cleavage assay is created for inhibitor validation assays [5]. Docking is the main tool for the computer-aided structural-based drug design—as can be seen, for example, in [6] and the references therein—sometimes followed by additional filtering of the ligand candidates using molecular dynamics (MD), quantum chemistry (QC) or something like intuition. Docking programs place ligands (candidates for inhibitors) in the active site of the target protein and calculates a score that characterizes the estimate of the free energy of protein–ligand binding. The lower this energy, the more likely it is that a given compound will show inhibitory activity in the experiments, and the more effective a drug based on such an inhibitor will be. There are many docking programs and each demonstrates a unique performance due to an individual combination of physical approximations, numerical methods and their program implementations [7,8]. The unbridled desire of developers of most docking programs to increase the speed of docking led to the use of extremely simplified models describing the interaction of ligands with a target protein. This led to unsatisfactory accuracy for most docking programs [9,10] and the need to use some additional tools after docking to select ligands for experimental testing using MD or quantum chemical calculations.

At the beginning of the pandemic, there were many publications devoted to the search for inhibitors of various coronavirus target proteins among approved drugs or among drugs in the last phases of clinical trials. In [8], the search for SARS-CoV-2 M${}^{pro}$ inhibitors was carried out in two libraries of the ZINC15 database [11]: in the FDA DrugBank for drugs approved in the USA [12] and in DrugBank 4.0 for drugs approved worldwide [13], a total of 7224 3D ligand structures. Virtual screening was performed using the SOL docking program [6] with subsequent quantum-chemical calculations of the protein–ligand binding enthalpy for ligands with the best docking scores. Finally, 21 compounds were selected as the best candidates for the role of M${}^{pro}$ inhibitors, but experimental trials have not yet been carried out. Most of these compounds were small molecules: derivatives of aminothiazole (Avatrombopag), piperazine (Blonanserin, Bromergocryptine, Buclizine), aminopyrimidine (Doxazosin, Nilotinib), etc.

In [14], Ghahremanpour et al. performed the docking of approximately two thousand approved drugs into M${}^{pro}$ of SARS-CoV-2. Docking was independently performed by four docking programs: Glide SP (Schrödinger, New York, NY, USA)(Scripps Research, La Jolla, CA, USA), and two methods implemented in AutoDock 4.2 content-type="color: #000000">(Scripps Research, La Jolla, CA, USA). The selection of ligands was carried out using the following consensus approach. For further processing, only ligands were selected that were in the top 10% according to three of the four docking protocols. This condition was met by 42 compounds, and further analysis using molecular dynamics, taking into account the chemical diversity of compounds and the ease of chemical synthesis of analogous compounds, led to the selection of 17 compounds for the experimental in vitro verification of their ability to inhibit M${}^{pro}$ in the FRET assay. As a result, four drugs were found (*manidipine*, *boceprevir*, *lercanidipine*, *bedaquiline*), for which the measured value of the inhibitor concentration, which suppresses the activity of M${}^{pro}$ by 50%, IC${}_{50}$, was in the range from 4.81 to 20.0 µM. Three of the four drugs contained a pyridine cycle in their structure. In [14], nothing was said about the suppression of coronavirus replication in cell culture by these inhibitors. Later, Zhang et al. [15] redesigned the weak M${}^{pro}$ inhibitor (*perampanel*)—as can be seen in Figure 1—the derivative of pyridine–2–one [14], using the free energy perturbation technique [16], and synthesized new compounds, several of which showed IC${}_{50}$≈ 20 nM, for the inhibition of M${}^{pro}$, and two of which suppressed the replication of SARS-CoV-2 in cell culture with EC${}_{50}$≈ 1 µM and have low cytotoxicity.

Shcherbakov et al. published [17] compounds containing a hydrogenated pyridine cycle bispidine derivatives, which have been proposed as inhibitors of SARS-CoV-2 M${}^{pro}$ based on the created pharmacophore model of the protease active site. Experimental testing confirmed the inhibitory activity of 14 bispidine compounds in the range of IC${}_{50}$ values 1–10 µM, and three compounds exhibited submicromolar activity (see one of them in Figure 2). The authors of [17] considered that these compounds form a covalent bond with Cys145. Almost all these bispidine inhibitors are non-toxic to the HEK293T cell line.

An example of pure experimental search for M${}^{pro}$ inhibitors among 10,755 compounds was presented by Zhu et al. in [18] where one M${}^{pro}$ inhibitor with IC${}_{50}$ = 0.26 µM and nine inhibitors with IC${}_{50}$ in the range of 2.63–13.00 µM were found, but only two of them (Valrycin B, Z-FA-FMK, see Figure 3a, Figure 3b, respectively) suppressed the virus replication in cell culture with submicromolar EC${}_{50}$. A similar attempt was undertaken within the framework of the E4C project [19], where an experimental (FRET and in cell culture) search for M${}^{pro}$ inhibitors was performed in three databases, with a total of 8702 compounds that are in preclinical and clinical trials. A dozen of M${}^{pro}$ inhibitors were found with IC${}_{50}$ in the range 0.1–7.4 µM, but only one of them *MG–132* is a peptide proteasome inhibitor, as can be seen in Figure 3c, which effectively suppressed the SARS-CoV-2 replication in a cell culture (EC${}_{50}$ = 0.1 µM).

The SARS-CoV-2 M${}^{pro}$ promising oral inhibitor belonging to the class of peptidomimetics, PF-07321332 (see Figure 4), was obtained by optimizing a previously found SARS-CoV-1 M${}^{pro}$ inhibitor. PF-07321332 exhibited the M${}^{pro}$ with the inhibition constant K${}_{i}$ = 3.11 nM, suppressed virus replication, EC${}_{50}$ = 85.3 nM in Vero E6 cell culture [20], exerted equipotent in vitro activity against the four SARS-CoV-2 variants of concern [21] and demonstrated oral activity in a mouse-adapted SARS-CoV-2 model. In December 2021, the FDA authorized PF-07321332 (INN: nirmatrelvir) as a first oral antiviral for the treatment of COVID-19 in combination with ritonavir. Pfizer also reported the development of an inhibitor of SARS-CoV-2 M${}^{pro}$ for the intravenous administration which possesses a chemotype similar to PF-07321332 [22].

In [23], Khan et al. carried out the virtual screening of 8000 compounds against M${}^{pro}$ using the MOE platform, among which there were 16 antiviral drugs acting on viral proteases, and an in-house database of natural and synthetic molecules. As a result of docking, the best 700 candidates were selected, and then their bound conformations were studied both visually and with using bioinformatics methods. As a result, two compounds from the in-house chemical library and three approved drugs (Saquinavir, Darunavir, Remdesivir) were accepted as promising candidates for the inhibition of the main protease of SARS-CoV-2, but there was no experimental confirmation. These drugs belong to the class of nucleotide analogues. Remdesivir, shown in Figure 5, has been found to suppress coronavirus replication in cell culture [24].

In [25], a docking study of 1,485,144 molecules, taken from the ChEMBL26 database, was performed by Tsuji et al. The docking was carried by subsequent use of the rDock [26] and AutoDock Vina docking programs. As a result, 29 of the best heterocyclic compounds (derivatives of aminothiazole, amino-1,3,5-triazine, aminopyrimidine, etc.) were selected as candidates for the role of inhibitors of SARS-CoV-2 M${}^{pro}$, but no experimental confirmation was reported. Using high-throughput screening, effective SARS-CoV-2 M${}^{pro}$ inhibitors were found in a library of 10,000 compounds, consisting of approved drugs, clinical-trial drug candidates, and natural products [27]. The most effective inhibitors proved to be Ebselen, Disulfiram, Tideglusib, Carmofur, Shikonin, and PX-12. All compounds (Figure 6) except Disulfiram are nitrogen-containing heterocycles (thiadiazoldione, pyrimidindione, imidazole derivatives). However, Ma et al. reported [28] that all these inhibitors were non-specific promiscuous cysteine modifiers and cannot be translated into real hits useful for drug design.

A review of promising inhibitors acting against SARS-CoV-2 and host targets was presented in [29], and recent discoveries of some inhibitors of SARS-CoV-2 M${}^{pro}$ were presented in [30,31,32,33]. In the latter, docking was used to screen 5903 drugs and investigational molecules against seven target enzymes known to be essential at different stages of the viral cycle. A whole series of works was devoted to the search for SARS-CoV-2 M${}^{pro}$ inhibitors in the series of quinoline derivatives [34,35].

In [36], Unoh et al. reported a promising non-covalent heterocyclic inhibitor, S-217622 (see Figure 7), discovered using a structure-based drug design (SBDD) strategy. The latter included THE docking of hundreds of thousands of compounds from the in-house library using the Glide docking program followed by the pharmacophore filtering applied to each docking pose, and the 300 top-scoring compounds were evaluated via enzymatic assays using mass spectrometry, giving some hit compounds with IC${}_{50}$< 10 µM. Further structure-based optimization of the hit compound resulted in the synthesis of S-217622 with IC${}_{50}$ = 0.013 µM, EC${}_{50}$ = 0.37 µM in SARS-CoV-2-infected Vero E6/TMPRSS2 cells. S-217622 exhibited similar antiviral activities against all tested SARS-CoV-2 variants, including the Omicron strain. At present S-217622 is in the phase 3 part of the phase 2/3 clinical trial. It should be noted that the inhibitor S-217622 is a hybrid molecule comprising cycles of 1,3,5-triazine, 1,2,4-triazole and indazole.

We see from this brief review that from the very beginning of the pandemic, computer modeling methods, and above all docking, began to be widely used in the search for antiviral substances. At the beginning, the search for SARS-CoV-2 M${}^{pro}$ inhibitors was carried out primarily, either among approved drugs, or drug compounds that, until recently, were in preclinical or clinical trials and were being developed for inhibiting therapeutic targets other than those associated with COVID-19. In this direction of drug development, despite the abundance of publications, experimental data were published for SARS-CoV-2 M${}^{pro}$ inhibitors with IC${}_{50}$ values mainly in the micromolar range. Some inhibitors had an IC${}_{50}$ of several tens or hundreds of nM, but most of these inhibitors binded covalently to M${}^{pro}$. Experiments showed that not all M${}^{pro}$ inhibitors suppressed the replication of the SARS-CoV-2 coronavirus in a cell culture, compounds that suppressed replication have EC${}_{50}$ values only in the micromolar range, and often have a low selectivity index (high cytotoxicity). However, a different approach to anti-coronavirus drug development has proven to be more effective. It was associated with a computer search for inhibitors in large databases of available drug-like compounds, the experimental confirmation of the activity of hits and their subsequent optimization and synthesis of new leader compounds. A prime example of the success of this approach is the SARS-CoV-2 non-covalent M${}^{pro}$ inhibitor S-217622, which is currently in clinical trials. We also note the success of nirmatrelvir, developed on the basis of the M${}^{pro}$ inhibitor of the old coronavirus SARS–CoV–1, which granted authorization by the FDA as a first oral antiviral for treatment of COVID-19.

In the present study, we used docking followed by quantum-chemistry post processing in the search for inhibitors of SARS-CoV-2 M${}^{pro}$. Initially, a database of low molecular weight drug-like compounds was processed, and 18 compounds were selected for experimental testing on the basis of their calculated binding characteristics. Some of these compounds inhibited SARS-CoV-2 replication in Vero E6 cell culture at the micromolar EC${}_{50}$ level. A search among hit analogs resulted in several compounds with micromolar EC${}_{50}$ values and low cytotoxicity. The new anti-SARS-CoV-2 compounds belong to three chemical classes.

## 2. Results and Discussion

### 2.1. Computer-Aided Initial Search for Inhibitors

Initially, the whole database of Voronezh State University containing approximately 19,000 ready-made drug-like low molecular weight compounds was screened using the SOL docking program. For each compound, several low-energy conformers were prepared, and as a result, more than 41,000 3D ligand conformers were docked. The SOL score −6.3 kcal/mol of the native compound, i.e., the X77 inhibitor crystallized with SARS-CoV-2 M${}^{pro}$ in the 6W63 PDB complex, was used as a threshold separating the predicted active from inactive compounds. Only those compounds that were more negative than the threshold score were selected for further processing, for which the population of the cluster with the lowest ligand energies was not less than 10. After docking, the best 1045 ligands satisfied these criteria, and for them, the enthalpy of protein–ligand binding was calculated. This value calculated for the native ligand X77 was equal to −58.5 kcal/mol, and this value was used as a criterion to filter out the worse candidates from 1045 ligands exhibiting good docking results. All calculations were performed on the Lomonosov–2 Moscow State University supercomputer [37].

Relying upon results of docking and quantum chemical calculations of the protein–ligand binding enthalpy, the analysis of the docked ligand poses in the protein active site, and providing the chemical diversity of the ligands, we selected a series of compounds for the first experimental testing [38]. All these ligands blocked the catalytic dyad of the SARS-CoV-2 M${}^{pro}$ in their docked conformations. Having obtained the experimental confirmation of the activity of some of the 18 tested compounds against SARS-CoV-2 in the cell culture, we selected the analogs of these active compounds for further experimental verification. These analogs were not included in the list of the best candidates for the first experiments, because their docked scores and/or binding enthalpies were not the best. In the following experiments, some of these analogs exhibited the suppression of SARS-CoV-2 replication in the cell culture.

### 2.2. Synthesis

Three classes of heterocyclic compounds were studied as promising inhibitors of the main protease of SARS-CoV-2. The general routes for the synthesis of the compounds under study are shown in Figure 1, Figure 2 and Figure 3.

The first class compounds (**3 a-b**) are 4,4-dimethyl-4,5-dihydro-1H-[1,2]dithiolo[3,4-c]quinoline-1-thiones **1 a-d** (Figure 1) N-acylated with naphthyloxyacetic acid chloride **2**. Compounds **1 a-d** were obtained according to known method [39,40,41].

The second class of compounds (**6 a-c**) are substituted into 4-phenyl-6-chloroquinolines obtained from N-(2-benzoyl-4-chlorophenyl)-2-chloroacetamide 4 (Figure 2). Compound 4 was obtained according to the known method [42]. Compound **6 a** was obtained in two stages. At the first stage, cyclization with morpholine in dry acetonitrile was carried out. At the second stage, O-alkylation was carried out with 2-chlorobenzyl chloride in the presence of cesium fluoride. Compounds **6 b-c** were obtained in one step by reaction 4 with piperidine-1-dithio acids in the presence of potassium acetate.

The third class of compounds (**9 a-c**) are substituted 2,4-diamino-1,3,5-triazines obtained from N,N${}^{\prime }$-diphenyl-6-chloro-1,3,5-triazine-2,4-diamine 7 by their reaction with cyclic amines **8 a-c** (Figure 2) according to the known method [43]. N,N${}^{\prime }$-diphenyl-6-chloro-1,3,5-triazine-2,4-diamine 7 was obtained according to the known method [44,45,46].

### 2.3. Protein–Ligand Binding: Modeling and *In Vitro* Testing

The calculated and measured characteristics of the three active compounds (**3a**, **6a**, **9a**) determined among the first set of 18 compounds selected for experimental testing [38] are presented in Table 1. These characteristics are: the estimate of the free energy of protein–ligand binding calculated by SOL (SOL score), the protein–ligand binding enthalpy $\Delta H_{bind}$, as well as the 50% effective concentrations (EC${}_{50}$) of the compounds that inhibit the viral replication the cell culture and the selectivity index $SI=CC_{50}/EC_{50}$. *Remdesivir* was used as a reference compound [47]. Table 1 does not show calculated values for *Remdesivir* because it binds to the RdRp target protein according to structural data obtained by electron microscopy [48]. *Remdesivir* does not bind to M${}^{pro}$.

We can see in Table 1 that, among the first 18 compounds which were tested in vitro, three compounds showed the suppression of the SARS-CoV-2 virus replication in the cell culture, and two of them, **3a** and **6a**, have the better or equal EC${}_{50}$ and SI values compared to *remdesivir*. As can be seen in Figure 8, these three compounds belong to three chemical classes: derivatives of [1,2]dithiolo[3,4-c]quinoline-1-thiones (**3a**), 1,3,5-triazine-2,4,6-triamines (**9a**) and 6-chloroquinoline (**6a**).

The next step in the selection of a candidate is the search for compounds whose molecules are the chemical analogs of the best ligands presented in Table 1. Several dozen analogs were selected and measured, and those compounds that demonstrated the suppression of SARS-CoV-2 replication in cell culture are presented in Table 2.

From Table 2, we can see that there are compounds suppressing SARS-CoV-2 replication in cell culture among the analogs of all three compounds in Table 1. Five of them have EC${}_{50}$< 10 µM. Three of these compounds, **9b**, **3b** and **3c** have better or equal EC${}_{50}$ values compared to *remdesivir* and two of these compounds, **3b** and **3c**, also have better or similar SI values compared to *remdesivir*. Structures of all compounds from Table 2 are presented in Figure 9.

Analyzing the structures depicted in Figure 8 and Figure 9, one can see that many identified inhibitors contain some reactive groups and thereby might possibly work via the covalent modification of M${}^{pro}$ or other viral enzymes. These include derivatives of dithioloquinoline (compounds **3a-d**) and carbodithioates (compounds **6b-c**). However, this estimation is somewhat speculative since these reactive groups cannot be found among classical warheads or in covalent binders from the literature (examples of warheads can be found in [49]). Their exact mechanism of action and selectivity will be determined in further iterations.

The docking position of the best ligand **3a** in the active site of M${}^{pro}$ is presented in Figure 10, respectively, and they are prepared by PyMol 2.5.0a0 (Schrödinger, New York, NY, USA) [50]. Interactions are predicted with PLIP [51].

According to its docking pose optimized with PM7, **3a** possesses good complementarity to the active site of M${}^{pro}$. Access to a catalytic dyad is clearly blocked by the dithioloquinoline core with dithiole ring participating in pi-stacking with *His41*. A naphthyl fragment is placed into a hydrophobic subpocket formed by *Met165*, *Leu167* and *Pro168*. A subpocket opened by the displacement of *Met49* is occupied with a methyl group. PLIP [51] predicts the presence H-bond between the amide oxygen of **3a** and a side chain of *Gln189*. To check the prediction, we performed semi-empirical calculations of energies for all H-bonds found for the **3a**–M${}^{pro}$ complex with the PM7 method using MOPAC. It turned out that the energy of this H-bond is less than 1.0 kcal/mol and its existence is not confirmed in semi-empirical calculations. The main reason for this may be due to the excessively long distance between the acceptor and a hydrogen atom—2.9 Å. Moreover, since a side chain of *Gln189* is quite flexible because of being placed toward the solvent, one can expect its low contribution to H-binding with ligands. Substituents in the benzene ring of the quinoline ring do not significantly affect both the calculated and experimental characteristics of compounds **3**. The similar analysis of the interactions of **6b** and **9b** compounds in their docking pose with M${}^{pro}$ is presented below.

In its docking pose, the **6b** compound forms T-shaped pi-stacking with *His41* by a phenyl ring placed towards the hydrophobic subpocket formed by *Met165*, *Leu167* and *Pro168*. The main difference between the predicted binding modes for **3a** and **6b** is that this hydrophobic subpocket is only partially occupied in the latter case. Thus, in theory, it can contribute to the difference in activity between these two compounds. Another interaction of **6b** found in docking studies includes one H-bond between a ligand aromatic amide group of pyridinone ring and backbone nitrogen of *Gly143*. According to the PM7 method, the energy of this H-bond is predicted to be −2.14 kcal/mol which is quite negative and can noticeably stabilize the observed geometry of the protein–ligand complex. The bent piperidine ring of the ligand occupies an additional crevice near *His163*.

A docking pose of **9b** reveals two adjacent H-bonds with backbone nitrogen and the oxygen of *Glu166*. According to the PM7 method, H-binding between *Glu166*-NH and the nitrogen atom, a triazine core of the ligand is very strong with the energy of −2.55 kcal/mol. The second H-bond involving the *Glu166*-O and aromatic amino group of the ligand is weaker and has an energy of −1.3 kcal/mol. One of two phenyl rings of **9b** occupies a hydrophobic subpocket mentioned above. In the comparison of a docking pose of **3a**, the **9b** ligand occupies an additional space near *His163* with the second phenyl group placed near this residue. The azepane ring is placed in a subpocket formed after the displacement of *Met49*.

In the present study, we used molecular modeling, a combination of the docking and quantum-chemical calculation of the protein–ligand binding enthalpy, in the search of the inhibitors of the SARS-CoV-2 main protease among almost twenty thousand compounds from Voronezh State University. As a result of the two-step search ten compounds of three chemical classes show a suppression of SARS-CoV-2 replication in cell culture. Four of these compounds have the EC${}_{50}$ values lower than 3 μM, which is better than the EC${}_{50}$ of *remdesivir* measured in our experiments. Two of these four compounds, **3a** and **3b**, have lower cytotoxicity, (SI > 86) than *remdesivir* (SI > 56). However, the molecular mechanism of action of these compounds on SARS-CoV-2 has yet to be proven. The docking results indicate that these compounds may possibly be inhibitors of the SARS-CoV-2 main protease, but this must be experimentally proven.

A priori, it is impossible to draw an unambiguous conclusion about the covalent or non-covalent binding of our inhibitors to the target protein. However, based on some of the following facts, we tend to assume non-covalent binding. The structure of our ligands **3** and **6** share common motifs with that of Ebselen, Disulfiram and PX-12. Ebselen (Figure 6a), like PX-12 (Figure 6f), is indeed with high probability a covalent protease inhibitor, which has been noted in a number of publications [27,52]. Covalent interaction indirectly confirms the non-specific action of Ebselen, Disulfirami, PX-12 and Tideglusib on the main protease. These inhibitors act on all cysteine proteases in a non-specific manner [28]. However, the same authors reported that Ebselen and other compounds can also inhibit M${}^{pro}$ via non-covalent binding [27]. The covalent binding of ligands to M${}^{pro}$, according to literature data, is due to the formation of bonds due to interaction with cysteine-145. However, the positions of our ligands, obtained by docking followed by quantum-chemical optimization, show that the convergence of the reactive groups of our ligands with cysteine-145 is difficult in space. For these positions, other non-covalent interactions were revealed. For example, the interaction of the dithiolthione ring of the **3a** compound with a fragment of histidine-41 or a carbonyl group with glutamine-189. That is, the spatial structure of the ligand and the binding site of the target suggest non-covalent binding. Certainly, it is the docking results, not the direct X-ray experimental confirmation. For all three types of ligands presented in the manuscript, we were unable to find information in the scientific literature on chemical interaction with compounds containing SH groups. The experimentally determined low cytotoxicity of our ligands **3, 6, 9** also speaks in favor of non-covalent interactions.

Several non-covalent inhibitors of the SARS-CoV-2 M${}^{pro}$ with the nanomolar IC${}_{50}$ values were found in [15], however, their EC${}_{50}$ values measured in the MTT test and the CC${}_{50}$ values in Vero E6 cell culture were worse than those of *remdesivir*. Thus, our best compounds inhibiting SARS-CoV-2 replication in cell culture are more active and less cytotoxic than the best compounds from [15].

Note that in the work [30], the best inhibitor of M${}^{pro}$ inhibits SARS-CoV-2 replication in the Vero E6 cell culture with only EC${}_{50}$ = 4.55 µM and CC${}_{50}$ > 20 µM, which is worse than the characteristics of our best compounds. In [31], several inhibitors of M${}^{pro}$ were identified with the micromolar IC${}_{50}$ values, but only one of them inhibited SARS-CoV-2 replication with EC${}_{50}$ = 0.32 µM and low cytotoxicity (SI = 120). Covalent inhibitors of SARS-CoV-2 M${}^{pro}$, *GC–373* and *GC–376*, with IC${}_{50}$ ≈ 0.3 µM, EC${}_{50}$ ≈ 1 µM and very low cytotoxicity CC${}_{50}$ > 200 µM were described in [53]. Then, two of their derivatives with improved properties (IC${}_{50}$ ≈ 0.07 µM and EC${}_{50}$ ≈ 0.7 µM) were presented in [32]. On the other hand, a non-covalent SARS-CoV-2 M${}^{pro}$ inhibitor with IC${}_{50}$ = 0.27 µM, EC${}_{50}$ = 1.27 µM and low cytotoxicity (CC${}_{50}$ > 100 µM) were published in [54]. Thus, with regard to the inhibition of SARS-CoV-2 replication in Vero E6 cell culture, our compounds **3a** and **3b** perform better than the best compounds from [54] and have lower cytotoxicity (higher SI value).

## 3. Conclusions

In this study, the results of the search for inhibitors of SARS-CoV-2 replication in a cell culture are presented. For the search, the atomistic model of the SARS-CoV-2 main protease was used, and those compounds were selected for in vitro testing which ligands bind to the active site of the main protease and block the catalytic dyad. The search was carried out in the database of drug-like low-molecular-weight compounds of the Department of Organic Chemistry of Voronezh State University, Russia. For molecular modeling, docking followed by quantum-chemical semi-empirical calculations is used. Docking is performed using the SOL docking program. For the ligands demonstrating the best values of the SOL score, the protein–ligand binding enthalpy is calculated using the PM7 quantum-chemical method with the COSMO solvent model. The ligands with the most negative values of the SOL score and the values of the binding enthalpy are selected for experimental testing in cell culture infected by SARS-CoV-2.

As a result of this study, ten inhibitors of SARS-CoV-2 replication in Vero E6 cells were experimentally confirmed with EC${}_{50}$ values in the range from 0.5 to 22.4 µM. These compounds belong to three different chemical classes: [1,2]dithiolo[3,4-c]quinoline-1-thione, 6-chloroquinoline, and 1,3,5-triazine-2,4,6-triamine. Three best inhibitors are derivatives of [1,2]dithiolo[3,4-c]quinoline-1-thione: **3a** (EC${}_{50}$ = 0.51 µM, SI > 412), **3b** (EC${}_{50}$ = 1.16 µM, SI > 86) and **3c** (EC${}_{50}$ = 2.71 µM, SI > 37) are no worse than best non-covalent inhibitors of SARS-CoV-2 replication published so far, and have lower cytotoxicity.

The results obtained open up an opportunity for the rational development of direct-acting antiviral drugs for the SARS-CoV-2 coronavirus in the fight against the COVID-19 pandemic.

## 4. Materials and Methods

### 4.1. The Target Protein Model

Currently, there are many high-quality complexes of SARS-CoV-2 M${}^{pro}$ with covalent and non-covalent inhibitors in the Protein Data Bank [55]. However, when we started our virtual screening campaign, one could find relatively few high-resolution structures for this target protein. At the time, only three complexes (PDB ID: 5R7Z, 5R83 and 6W63) were co-crystallized with non-covalent inhibitors. They had no missed residues or atoms, and their resolution was better than 2.2 Å. The atomistic models of M${}^{pro}$ were prepared using these structures (details can be found in [38]) and the docking of native ligands was performed using the SOL docking program [6,56]. The docking of the native ligand into the M${}^{pro}$ model constructed using the 5R7Z structure was unsuccessful: the docked and crystallized native ligand poses differed by RMSD = 7 Å. Native docking into M${}^{pro}$ models constructed using 5R83 and 6W63 were successful with RMSD = 1.19 and 1.31 Å, respectively. Comparison of these structures revealed the mobility of the *Met49* residue in the active site. For further work we chose the model constructed using the 6W63 complex of M${}^{pro}$ with the X77 inhibitor containing 7 torsions because it had a more open active site. If we compare the 6W63 complex with new complexes of M${}^{pro}$ with non-covalent inhibitors recently deposited in PDB, we cannot see any dramatic advantages for these complexes over the model based on 6W63.

### 4.2. The Database of Organic Compounds

The search for M${}^{pro}$ inhibitors was carried out in the database of the Department of Organic Chemistry of Voronezh State University, which contains approximately 19,000 ready-made compounds [57]. The database contains a wide spectrum of nitogen-, oxygen and sulfur-containing heterocyclic compounds. Among them are hydroquinoline derivatives with antibacterial, antifungal, anticoagulant activity [40,58,59,60,61]; aminopyrimidines and pyrrolo[3,2,1-ij]quinolin-2-ones, which are factor Xa and protein kinases inhibitors [62,63]; and various plant growth stimulants of the getarylcarboxylic acid class [64,65]. Each ligand in the database is represented as a 2D structure. In the course of the 2D-to-3D transformation, several low-energy conformers are kept. These are different conformers of non-aromatic cycles and macrocycles. The low-energy conformers of ligands are obtained by the OpenBabel software [66]. Ligand protonation is carried out using the ChemAxon pKa Plugin [67] at pH = 7.4. Overall, more than 41,000 3D ligand structures are prepared for docking.

### 4.3. Docking and Postprocessing

Docking is performed by the SOL program [6,7,8,56,68,69] which has been successfully used over the past 15 years to develop the inhibitors of some target proteins: thrombin (coagulation factor IIa) [70], urokinase (uPA) [71,72], coagulation factors Xa [58,63] and XIa [60]. The performance of SOL is based on the docking paradigm which assumes that a ligand binds near the global minimum of the energy of the protein–ligand complex. SOL uses the MMFF94 force field, a preliminary calculated grid of potentials describing the Coulomb and van der Waals interaction of ligand atoms with the protein, and taking into account the desolvation effect. The latter is described by the difference of hydration energy of the protein–ligand complex and the sum of hydration energies of the unbound protein and the unbound ligand. For the calculation of the desolvation energy, a simplified generalized Born model is used [73]. The genetic global optimization algorithm is used for ligand positioning in the active site of the target protein. Specific features of SOL are large parameters of the genetic algorithm such as the population size (30,000), number of generations (1000), and 50 independent runs of the algorithm. After fifty independent runs, fifty corresponding positions of the ligand with the lowest energies were clustered using the RMSD = 1 Å criterion between the positions of the ligand in the cluster. The ligand poses with the lowest energies, and by definition, belong to the first cluster. Only those docking solutions are considered reliable if the population of the first cluster is greater than or equal to 10. The high population of the first cluster indicates the convergence of independent runs of the genetic algorithm to the unique global minimum. The native docking of the X77 ligand into the corresponding protein of the 6W63 complex gives the SOL score −6.3 kcal/mol, and we use this value as a threshold separating good and bad binders (inhibitors).

In our virtual screening using docking, we assumed a non-covalent mechanism of protein–ligand binding. Chemical reactions between a target protein and ligand were not considered. Nevertheless, we assume that docking can also be useful for the search for covalent inhibitors. Such ligands at the initial phase of interaction with the protein must first bind to it in the desired orientation due to intermolecular interactions, which are taken into account during docking. It is from this position of the ligand that the further formation of a covalent bond between the protein and the ligand by a chemical reaction can occur if the ligand has an appropriate chemical warhead.

The best ligands with an SOL score which was more negative than the threshold were subjected to quantum-chemical post-processing. The goal of post-processing is to additionally select the best candidates for experimental verification. For this purpose, the protein–ligand binding enthalpy is calculated as follows. Calculations were carried out using the MOPAC program [74] by the PM7 quantum-chemical semiempirical method [75] with the COSMO implicit solvent model [76]; for brevity, we denote such calculations as PM7/COSMO. The PM7/COSMO energy of the protein–ligand complex was optimized, starting from the best position of the ligand found during docking, with the varying coordinates of all ligand atoms. The PM7+COSMO energy of the unbound protein was calculated for the conformation used for docking, and for the unbound ligand, the energy of the ligand conformation with the lowest energy calculated by the PM7+COSMO method was used. The enthalpy of binding was calculated as the difference between these three PM7/COSMO energies: the energies of the unbound protein and unbound ligand were subtracted from the energy of the complex [38]. The MOPAC program and the PM7 method were chosen for these calculations because MOPAC could process protein–ligand complexes with sufficient speed due to the use of the localized molecular orbital method implemented in the MOZYME module, and PM7 described dispersion interactions as well as hydrogen and halogen bonds with an accuracy approaching that obtained by DFT methods. The combination of PM7/COSMO gave a strong correlation coefficient (0.74) between the calculated and experimentally measured protein–ligand binding enthalpy [77,78].

### 4.4. Antiviral Activity (Wild-Type SARS-CoV-2)

We seeded Vero E6 cells onto a white, flat bottom 96-well plate at 10,000 cells per well in Dulbecco’s modified Eagle’s medium supplemented with 10% heat-inactivated fetal bovine serum, 100 U/mL penicillin–streptomycin (Life Technologies, Carlsbad, CA, USA), and buffered with 10 mM HEPES and incubated in a humidified 5% CO2 incubator at 37 °C. The media were removed 24 h later and replaced with the 50 µL/well of the same media but containing 2% heat-inactivated fetal bovine serum (D-2) instead of 10%. In D-2, compounds were serially diluted 5-fold in a 5-point series and added in a 25 µL volume to the 96-well plate to make final concentrations ranging from 100 µM to 32 nM on the assay plate for 2 h. We then infected the cells (with compounds still on them) with 25 µL/well of SARS-CoV-2 at an MOI of 0.1, with a final total volume of 100 µL/well. We incubated the plates for 96 h as described above. Compound cytotoxicity was determined using the same assay, but instead of adding virus, we added 25 µL of D-2.

The number of viable cells (protected from the cytopathic action of the virus) was determined using 3-(4,5-dimethylthiazol-2-Yl)-2,5-diphenyl-2H-tetrazoliumbromide (MTT method). The test is based on the transformation of pale yellow 3-(4,5-dimethylthiazoline-2)-2,5-diphenyltetrazolium bromide (Sigma-Aldrich, St. Louis, MO, USA) into violet formazan under the action of the enzyme succinate dehydrogenase. To do this, MTT solution (50 µL, 5 mg/mL) was added to the cell supernatant. The cell plate was incubated for 90 min at 37 °C. The supernatant was removed, and the cells were fixed with 4% formaldehyde for 30 min. Tetrazolium crystals were dissolved in 1 mL of 96% ethanol for 10 min.

On a multifunctional spectrophotometer xMark (Bio-Rad, Hercules, CA, USA), in 96-well plates at a wavelength of 450 nm, measure the optical density (OD), which correlates with the number of viable cells in a monolayer, protected from the cytopathic effect (CPE) of the virus or the cytotoxic action of the test compound.

The OD values are used to determine the 50% cytotoxic concentration (CC${}_{50}$) and 50% inhibitory (effective) concentration (EC${}_{50}$) of the compound using the SoftMaxPro-4.0 (Molecular Devices Co., Sunnyvale, CA). The CC${}_{50}$ value is the concentration of the compound at which 50% of the cells in an uninfected monolayer are destroyed (lose their viability) due to the toxicity of the compound. The EC${}_{50}$ value is the concentration of the compound at which 50% of the cells in the infected monolayer are not destroyed (remain viable), and is an indicator of the effectiveness of suppressing viral replication. The selectivity index (SI) of the test compound is calculated: SI = CC${}_{50}$/EC${}_{50}$. SI is the very first indicator showing the promising nature of this compound as a basis for a future drug. If SI ≫ 1, the therapeutic concentration of the compound, i.e., the concentration of the compound effectively suppressing the virus replication, is much less than the concentration of this compound causing a cytotoxic effect in cell culture. The existing drug *remdesivir* [47] was used as a reference to measure the inhibition of the replication of SARS-CoV-2 coronavirus in cell culture.

### 4.5. Chemistry

#### 4.5.1. General

Melting points were determined on a PTP-M apparatus. The 1H and 13C NMR spectra were recorded on a Bruker DRX-500 spectrometer in DMSO-d${}_{6}$ at 500 and 125 MHz, respectively. TMS was used as the internal standard. HPLC–HRMS analyses were performed on an Agilent Infinity 1260 liquid chromatograph equipped with an Agilent 6230 TOF mass selective detector. The conditions of chromatographic separation were the following: mobile phase 0.1% formic acid in CH${}_{3}$CN (eluent A)/0.1% formic acid in water (eluent B), gradient 0–100%: A, 3.5 min, 50%; A, 1.5 min, 50–100%; B, 3.5 min, 50%; B, 1.5 min, 50–0%; flow rate 0.4 mL/min, column–Poroshell 120 EC-C18 (4.6 50 mm, 2.7 mm), thermostat at 28, electrospray ionization (ESI, capillary voltage 3.5 kV; fragmentor voltage +191 V; OctRF +66 V–positive polarity). The reactions were monitored and the purity of the products was verified by TLC with Merck TLC Silica gel 60 F${}_{254}$ plates using chloroform as eluent. The solvents were purified according to standard methods. Commercially available reagents from Lancaster were used in the syntheses.

#### 4.5.2. General Procedure for Synthesis of Substituted

##### 4,4-dimethyl-5-[(naphthalen-2-yloxy)acetyl]-7-R2-8-R1-4,5-dihydro-1H-[1,2]dithiolo[3,4-c]quinoline-1-thiones **3 a-d**

A mixture of 4,4-dimethyl-7-R2-8-R1-4,5-dihydro-1H-[1,2]dithiolo[3,4-c]quinoline-1-thione 1 (5 mmol) and (2-naphthyloxy)acetyl chloride (6 mmol) in dry toluene (45 ml) and pyridine (5 ml) was refluxed for 5 h. Pyridinium chloride was filtered. Solvent was distilled off under reduced pressure, the precipitate was filtered and recrystallized from toluene to furnish the desired products **3 a-d**.

##### 4,4,7,8-tetramethyl-5-[(naphthalen-2-yloxy)acetyl-4,5-dihydro-1H-[1,2]dithiolo[3,4-c]quinoline-1-thione **3 a**

Orange powder (yield 1.91 g, 80%), m.p. = 189–190 °C; ${}^{1}$H NMR (DMSO-d6, 500 MHz) d ppm: 1.64 (br s, 6H, (CH${}_{3}$)${}_{2}$), 2.23 (s, 3H, CH${}_{3}$), 2.24 (s, 3H, CH${}_{3}$), 4.81 (s, 2H, CH${}_{2}$CO), 6.76 (s, 1H, H-6 quinoline), 6.77–6.85 (m, 1H, H-Ar), 7.28–7.31 (m, 1H, H-Ar), 7.32–7.40 (m, 2H, H-Ar), 7.57 (d, *J* = 8.1 Hz, 1H, H-Ar), 7.65 (d, *J* = 8.9 Hz, 1H, H-Ar), 7.74 (d, *J* = 8.1 Hz, 1H, H-Ar), 8.93 (s, 1H, H-9 quinoline); ${}^{13}$C NMR (DMSO-d6, 125 MHz) d ppm: 19.2, 25.5, 61.2, 67.4, 106.5, 118.0, 122.8, 123.7, 125.7, 126.3, 126.4, 127.3, 128.5, 129.4, 133.0, 133.6, 134.0, 134.4, 138.0, 155.0, 169.8, 178.9, 210.9; HPLC-HRMS (ESI) calculated for C${}_{26}$H${}_{23}$NO${}_{2}$S${}_{3}$ + H${}^{+}$, 478.0965; found, 478.0964 (see Appendix A).

##### 4,4,7-trimethyl-5-[(naphthalen-2-yloxy)acetyl-4,5-dihydro-1H-[1,2]dithiolo[3,4-c]quinoline-1-thione **3 b**

Orange powder (yield 1.78 g, 77%), m.p. > 144–145 °C; ${}^{1}$H NMR (DMSO-d6, 500 MHz) d ppm: 1.65 (br s, 6H, (CH${}_{3}$)${}_{2}$), 2.32 (s, 3H, CH${}_{3}$), 4.84 (s, 2H, CH${}_{2}$CO), 6.77–6.80 (m, 2H, H-Ar, H-quinoline), 7.20–7.40 (m, 4H, 3H-Ar, H-quinoline), 7.58 (d, *J* = 8.1 Hz, 1H, H-Ar), 7.60–7.64 (m, 1H, H-Ar), 7.75 (d, *J* = 8.1 Hz, 1H, H-Ar), 9.02 (d, *J* = 8.1 Hz, 1H, H-9 quinoline); ${}^{13}$C NMR (DMSO-d6, 125 MHz) d ppm: 20.8, 25.5, 61.2, 67.5, 106.5, 118.1, 122.6, 122.8, 123.8, 125.2, 126.4, 126.6, 127.4, 128.6, 129.6, 133.6, 134.4, 135.2, 139.5, 155.0, 170.0, 178.8, 211.0; HPLC-HRMS (ESI) calculated for C${}_{25}$H${}_{21}$NO${}_{2}$S${}_{3}$ + H${}^{+}$, 464.0808; found, 464.0812.

##### 4,4,8-trimethyl-5-[(naphthalen-2-yloxy)acetyl-4,5-dihydro-1H-[1,2]dithiolo[3,4-c]quinoline-1-thione **3 c**

Orange powder (yield 1.86 g, 80%), m.p. = 153–154 °C; ${}^{1}$H NMR (DMSO-d6, 500 MHz) d ppm: 1.65 (br s, 6H, (CH${}_{3}$)${}_{2}$)), 2.35 (s, 3H, CH${}_{3}$), 4.79 (s, 2H, CH${}_{2}$CO), 6.77–6.83 (m, 2H, H-Ar, H-quinoline), 7.24–7.40 (m, 3H, 2H-Ar, H-quinoline), 7.47 (d, *J* = 8.0 Hz, 1H, H-Ar), 7.58 (d, *J* = 8.1 Hz, 1H, H-Ar), 7.67 (d, *J* = 8.9 Hz, 1H, H-Ar), 7.75 (d, *J* = 8.0 Hz, 1H, H-Ar), 8.99 (s, 1H, H-9 quinoline); ${}^{13}$C NMR (DMSO-d6, 125 MHz) d ppm: 20.9, 25.5, 61.2, 67.2, 106.4, 118.0, 123.3, 123.7, 124.8, 125.1, 126.3, 127.3, 128.5, 129.4, 129.8, 132.7, 133.6, 134.2, 135.5, 155.0, 169.7, 179.9, 211.0; HPLC-HRMS (ESI) calculated for C${}_{25}$H${}_{21}$NO${}_{2}$S${}_{3}$ + H${}^{+}$, 464.0808; found, 464.0811

##### 8-methoxy-4,4-dimethyl-5-[(naphthalen-2-yloxy)acetyl-4,5-dihydro-1H-[1,2]dithiolo[3,4-c]quinoline-1-thione **3 d**

Orange powder (yield 1.99 g, 83%), m.p. = 136–137 °C; ${}^{1}$H NMR (DMSO-d6, 500 MHz) d ppm: 1.32 (br s, 3H, CH${}_{3}$), 2.04 (br s, 3H, CH${}_{3}$), 3.79 (s, 3H, OCH${}_{3}$), 4.77 (s, 2H, CH${}_{2}$CO), 6.74 (s, 1H, H-6 quinoline) (m, 1H, H-Ar), 6.82–6.85 (m, 1H, H-Ar), ), 7.00–7.03 (m, 1H, H-Ar), 7.28-7.40 (m, 2H, 2H-Ar), 7.54–7.59 (m, 2H, H-Ar, H-quinoline), 7.67 (d, *J* = 8.9 Hz, 1H, H-Ar), 7.75 (d, *J* = 8.0 Hz, 1H, H-Ar), 8.81 (s, 1H, H-9 quinoline); ${}^{13}$C NMR (DMSO-d6, 125 MHz) d ppm: 25.8, 55.4, 61.3, 67.2, 106.4, 108.8, 114.3, 118.1, 123.8, 126.3, 126.4, 126.5, 127.4, 128.0, 128.5, 129.5, 133.6, 134.1, 155.1, 157.1, 169.7, 180.7, 211.0; HPLC-HRMS (ESI) calculated for C${}_{25}$H${}_{21}$NO${}_{3}$S${}_{3}$ + H${}^{+}$, 480.0758; found, 480.0755.

#### 4.5.3. Synthesis of Substituted 6-Chloro-4-phenylquinolines **6 a-c**

##### 6-chloro-2-[(2-chlorobenzyl)oxy]-3-(morpholin-4-yl)-4-phenylquinoline **6 a**

1 Step: A mixture of N-(2-benzoyl-4-chlorophenyl)-2-chloroacetamide 4 (5 mmol) and morpholine (15 mmol) in dry acetonitrile (50 mL) was refluxed for 3 h. The reaction mixture was poured into water, and the precipitate was filtered and recrystallized from dioxan to furnish the desired 6-chloro-3-(morpholin-4-yl)-4-phenylquinolin-2(1H)-one 5 (yield 1.44 g, 85%).

2 Step: To the mixture of 6-chloro-3-(morpholin-4-yl)-4-phenylquinolin-2(1H)-one 5 (5 mmol) and cesium chloride (5 mmol) in dry DMF (50 mL) was added dropwise with stirring 2-chlorobenzylchloride and refluxed for 4 h. The reaction mixture was poured into water, and the precipitate was filtered. The product was purified by flash chromatography. A mixture of ethyl acetate: hexane in a volume ratio of 3:1 was used as elution solution.

White powder (yield 1.74 g, 75%), m.p. = 178–179 °C; ${}^{1}$H NMR (DMSO-d6, 500 MHz) d ppm: 2.81 (s, 4H, 2CH${}_{2}$ morpholine), 3.32 (s, 4H, 2CH${}_{2}$ morpholine), 5.58 (s, 2H, CH${}_{2}$Ar), 6.71 (d, *J* = 7.5 Hz, 1H, H-Ar), 7.04–7.06 (m, 1H, H-Ar), 7.17–7.20 (m, 1H, H-Ar), 7.23–7.26 (m, 1H, H-Ar), 7.33 (t, *J* = 7.5 Hz, 1H, H-Ar), 7.40 (d, *J* = 7.3 Hz, 2H, 2H-Ar), 7.43-7.46 (m, 1H, H-Ar), 7.51–7.55 (m, 1H, H-Ar), 7.57–7.62 (m, 3H, 3H-Ar); ${}^{13}$C NMR (DMSO-d6, 125 MHz) d ppm: 44.1, 50.2, 66.5, 116.7, 123.0, 125.1, 126.4, 126.6, 127.6, 128.1, 128.4, 128.5, 128.9, 129.6, 129.7, 131.8, 133.2, 135.1, 135.2, 138.7, 139.4, 159.8; HPLC-HRMS (ESI) calculated for C${}_{26}$H${}_{22}$C${}_{12}$N${}_{2}$O${}_{2}$ + H${}^{+}$, 465.1132; found, 465.1126.

##### 6-chloro-2-oxo-4-phenyl-1,2-dihydroquinolin-3-yl piperidine-1-carboditioate **6 b**

Piperidine-1-yldithiocarboxylic acid (5 mmol) was added dropwise and stirred into the mixture of N-(2-benzoyl-4-chlorophenyl)-2-chloroacetamide 4 (5 mmol) and potassium acetate (15 mmol) in dry DMF (50 mL) and then at 80 °C for 6 h. The reaction mixture was poured into water, and the precipitate was filtered and recrystallized from dioxane.

Yellow powder (yield 1.61 g, 78%), m.p. = 261–262 °C; ${}^{1}$H NMR (DMSO-d6, 500 MHz) d ppm: 1.43–1.59 (m, 2H, CH${}_{2}$ piperidine), 3.76–3.78 (m, 4H, 2CH${}_{2}$ piperidine), 4.04–4.10 (m, 4H, 2CH${}_{2}$ piperidine), 6.82–6.83 (m, 1H, 1H-Ar), 7.30–7.33 (m, 2H, 2H-Ar), 7.40 (d, *J* = 8.8 Hz, 1H, 1H-Ar), 7.43–7.50 (m, 3H, 3H-Ar), 7.60–7.63 (m, 1H, H-Ar), 12.24 (s, 1H, NH); ${}^{13}$C NMR (DMSO-d6, 125 MHz) d ppm: 23.3, 25.0, 25.7, 52.1, 117.4, 120.9, 125.7, 126.1, 127.8, 128.2, 128.6, 131.3, 135.6, 138.1, 155.5, 159.0, 192.0; HPLC-HRMS (ESI) calculated for C${}_{21}$H${}_{19}$ClN${}_{2}$OS${}_{2}$ + H${}^{+}$, 415.0701; found, 415.0719.

##### 6-chloro-2-oxo-4-phenyl-1,2-dihydroquinolin-3-yl 2-methylpiperidine-1-carboditioate **6 c**

2-methylpiperidine-1-yldithiocarboxylic acid (5 mmol) was added dropwise and stirred into the mixture of N-(2-benzoyl-4-chlorophenyl)-2-chloroacetamide 4 (5 mmol) and potassium acetate (15 mmol) in dry DMF (50 mL) and then at 80 °C for 6 h. The reaction mixture was poured into water, and the precipitate was filtered and recrystallized from dioxane.

White powder (yield 1.60 g, 75%), m.p. = 265–266 °C; ${}^{1}$H NMR (DMSO-d6, 500 MHz) d ppm: 1.11–1.12 (m, 3H, CH${}_{3}$), 1.30–1.64 (m, 4H, CH piperidine), 3.05–3.20 (m, 1H, CH piperidine), 4.12–4.22 (m, 1H, CH piperidine), 4.59–4.63 (m, 1H, CH piperidine), 5.01–5.20 (m, 1H, CH piperidine), 5.49–5.61 (m, 1H, CH piperidine), 6.82–6.83 (m, 1H, 1H-Ar), 7.30–7.33 (m, 2H, 2H-Ar), 7.40 (d, *J* = 8.8 Hz, 1H, 1H-Ar), 7.43–7.49 (m, 3H, 3H-Ar), 7.60–7.63 (m, 1H, H-Ar), 12.23 (s, 1H, NH); ${}^{13}$C NMR (DMSO-d6, 125 MHz) d ppm: 15.5, 17.7, 24.9, 29.7, 45.9, 53.9, 54.8, 117.5, 121.0, 125.7, 126.1, 127.7, 128.2, 128.6, 131.3, 135.6, 138.1, 155.5, 159.1, 192.1; HPLC-HRMS (ESI) calculated for C${}_{22}$H${}_{21}$ClN${}_{2}$OS${}_{2}$ + H${}^{+}$, 429.0858; found, 429.0872.

#### 4.5.4. General Procedure for Synthesis of Substituted

##### 6-R-N,N${}^{\prime }$-diaryl-1,3,5-triazine-2,4-diamine **9 a-c**

Cyclic amine 8 (5.25 mmol) and potassium carbonate (5.5 mmol) in dry acetonitrile (50 mL) were added to the mixture of 6-chloro-N,N${}^{\prime }$-diaryl-1,3,5-triazine-2,4-diamine 7 (5 mmol) which were then heated at 80 °C for 3 h. The reaction mixture was poured into water, and the precipitate was filtered and recrystallized from dioxane to furnish the desired products **9 a-c**.

##### 6-(azepan-1-yl)-N,N${}^{\prime }$-di(naphthalen-2-yl)-1,3,5-triazine-2,4-diamine **9 a**

White powder (yield 2.14 g, 93%), m.p. = 164–165 °C; ${}^{1}$H NMR (DMSO-d6, 500 MHz) d ppm: 1.44–1.49 (m, 4H, 2CH${}_{2}$ azepan), 1.52–1.61 (m, 4H, 2CH${}_{3}$ azepan), 3.51–3.54 (m, 4H, 2CH${}_{3}$ azepan), 7.37 (t, *J* = 7.7 Hz, 2H, 2H-Ar), 7.47–7.50 (m, 4H, 4H-Ar), 7.61–7.67 (m, 4H, 4H-Ar), 7.86–7.90 (m, 2H, 2H-Ar), 8.01–8.05 (m, 2H, 2H-Ar), 8.91 (s, 2H, 2NH); ${}^{13}$C NMR (DMSO-d6, 125 MHz) d ppm: 26.5, 27.5, 45.7, 122.1, 123.3, 124.1, 125.2, 125.3, 125.5, 127.8, 128.6, 133.6, 135.0, 164.7, 165.6; HPLC-HRMS (ESI) calculated for C${}_{29}$H${}_{28}$N${}_{6}$ + H${}^{+}$, 461.2449; found, 461.2468.

##### 6-(azepan-1-yl)-N,N’-diphenyl-1,3,5-triazine-2,4-diamine **9 b**

White powder (yield 1.66 g, 92%), m.p. = 160–161 °C; ${}^{1}$H NMR (DMSO-d6, 500 MHz) d ppm: 1.45–1.50 (m, 4H, 2CH${}_{2}$ azepan), 1.70–1.74 (m, 4H, 2CH${}_{2}$ azepan), 3.66–3.71 (m, 4H, 2CH${}_{2}$ azepan), 6.91 (t, *J* = 7.4 Hz, 2H, 2H-Ar), 7.52 (t, *J* = 7.4 Hz, 4H, 4H-Ar), 7.76 (d, *J* = 7.9 Hz, 4H, 4H-Ar), 9.05 (s, 2H, 2NH); ${}^{13}$C NMR (DMSO-d6, 125 MHz) d ppm: 26.5, 27.4, 46.1, 119.6, 121.4, 128.2, 140.3, 163.9, 164.4; HPLC-HRMS (ESI) calculated for C${}_{21}$H${}_{24}$N${}_{6}$ + H${}^{+}$, 361,2136; found, 361.2161.

##### 6-(4-methylpiperazin-1-yl)-N,N${}^{\prime }$-di(naphthalen-1-yl)-1,3,5-triazine-2,4-amine **9 c**

White powder (yield 2.05 g, 89%), m.p. = 160–161 ${}^{{}^{\circ}}$C; ${}^{1}$H NMR (DMSO-d6, 500 MHz) d ppm: 2.6 (s, 3H, CH${}_{3}$), 2.22–2.26 (m, 4H, 2CH${}_{2}$ piperazine), 3.52–3.61 (m, 4H, 2CH${}_{2}$ piperazine), 7.35 (t, *J* = 7.8 Hz, 2H, 2H-Ar), 7.44–7.50 (m, 4H, 4H-Ar), 7.59 (d, *J* = 7.4 Hz, 2H, 2H-Ar), 7.65 (d, *J* = 8.1 Hz, 2H, 2H-Ar), 7.87–7.89 (m, 2H, 2H-Ar), 8.01–8.04 (m, 2H, 2H-Ar), 8.92 (s, 2H, 2NH); ${}^{13}$C NMR (DMSO-d6, 125 MHz) d ppm: 42.4, 45.8, 54.4, 122.4, 123.4, 124.4, 125.3, 125.4, 125.7, 127/9, 128.7, 129.4, 133.7, 134.9, 164.8, 165.7; HPLC-HRMS (ESI) calculated for C${}_{28}$H${}_{27}$N${}_{7}$ + H${}^{+}$, 462,2402; found, 462.2409.

## Figures and Tables

**Figure 1 molecules-27-05732-f001:**
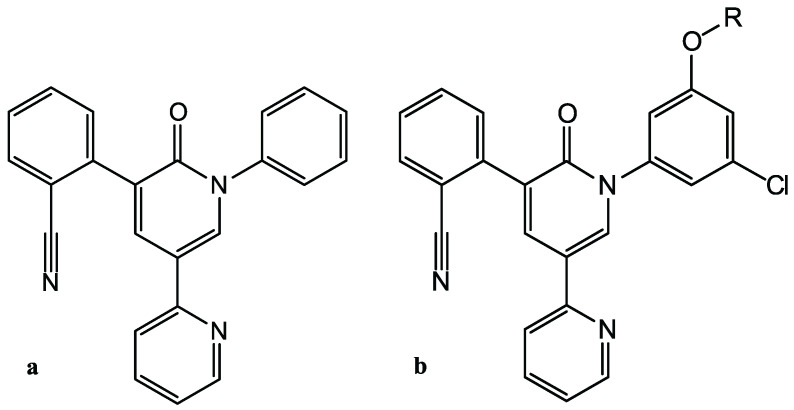
SARS-CoV-2 M${}^{pro}$ inhibitors: (**a**): Perampanel; and (**b**): Perampanel derivatives after structure optimization.

**Figure 2 molecules-27-05732-f002:**
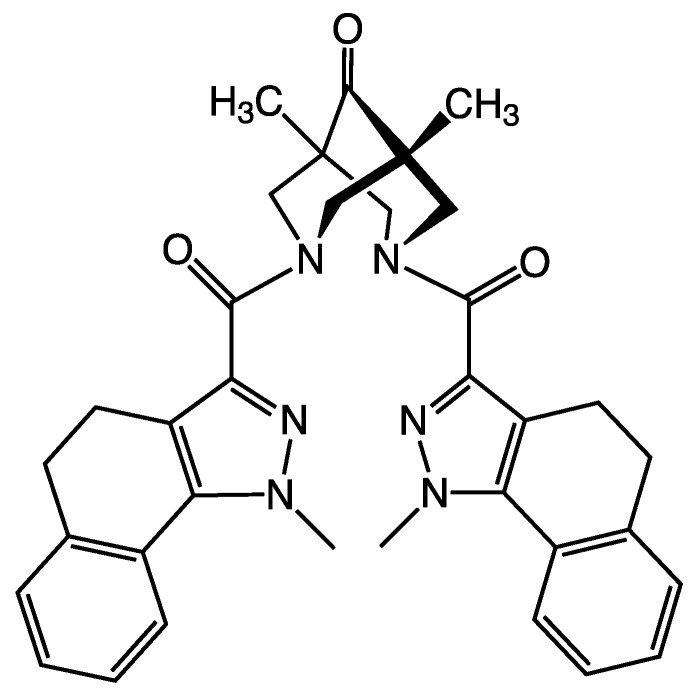
SARS-CoV-2 M${}^{pro}$ inhibitor Bispidine derivative.

**Figure 3 molecules-27-05732-f003:**
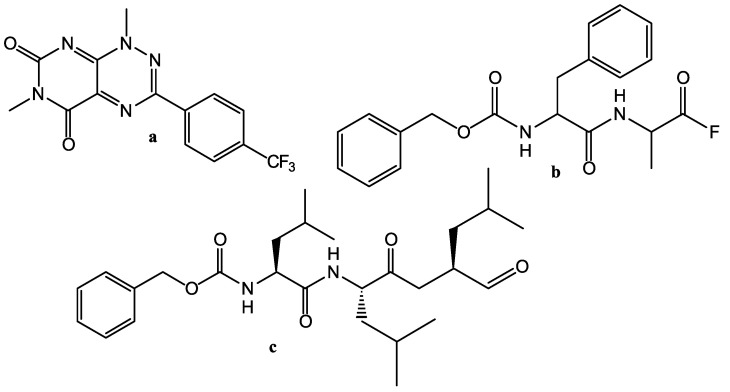
SARS-CoV-2 M${}^{pro}$ inhibitors: (**a**): Valrycin B; (**b**): Z-FA-FMK; and (**c**): **MG–132**.

**Figure 4 molecules-27-05732-f004:**
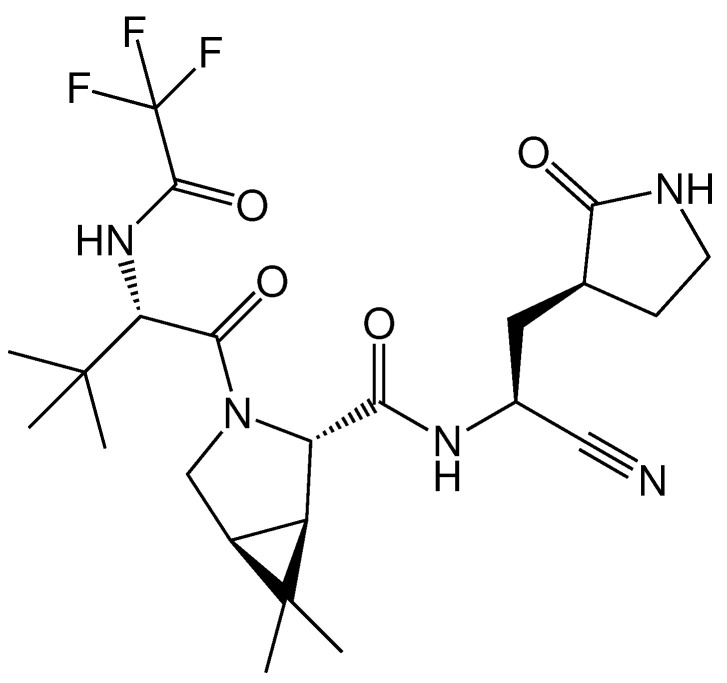
SARS-CoV-2 M${}^{pro}$ inhibitor PF-07321332 (nirmatrelvir), which was granted authorization by FDA in December 2021.

**Figure 5 molecules-27-05732-f005:**
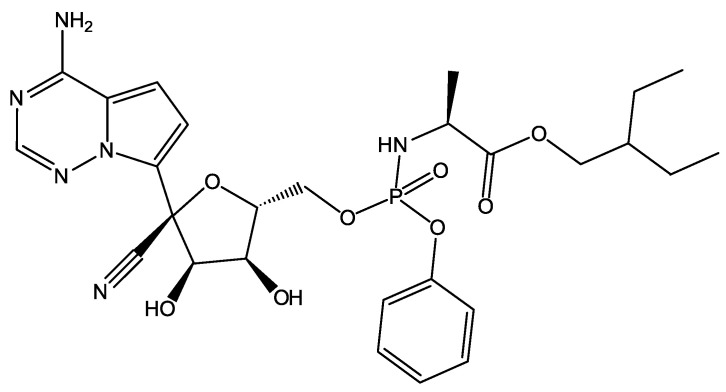
Remdesivir is an inhibitor of the SARS-CoV-2 M${}^{pro}$.

**Figure 6 molecules-27-05732-f006:**
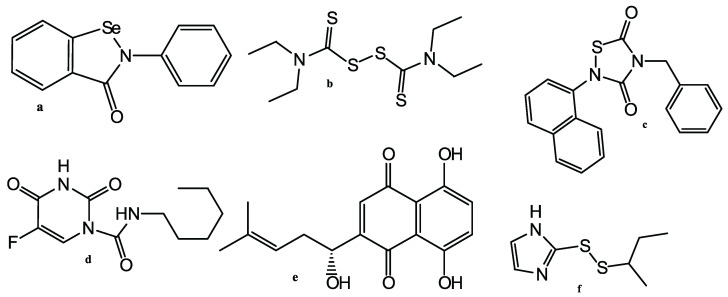
Non-specific promiscuous SARS-CoV-2 M${}^{pro}$ inhibitors: (**a**): Ebselen; (**b**): Disulfiram; (**c**): Tideglusib; (**d**): Carmofur; (**e**): Shikonin; and (**f**): PX-12.

**Figure 7 molecules-27-05732-f007:**
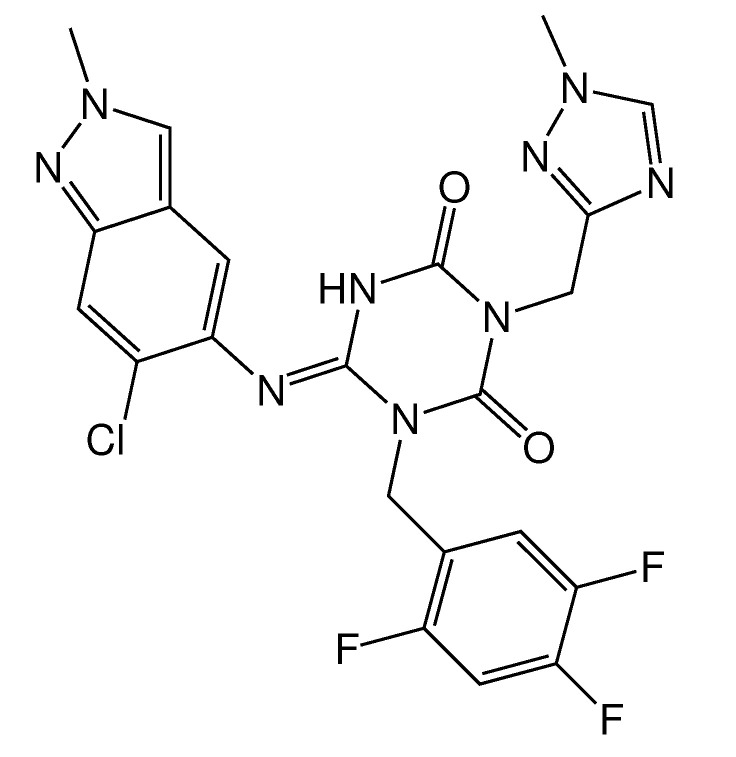
SARS-CoV-2 M${}^{pro}$ inhibitor S-217622, which is under clinical development.

**Scheme 1 molecules-27-05732-sch001:**
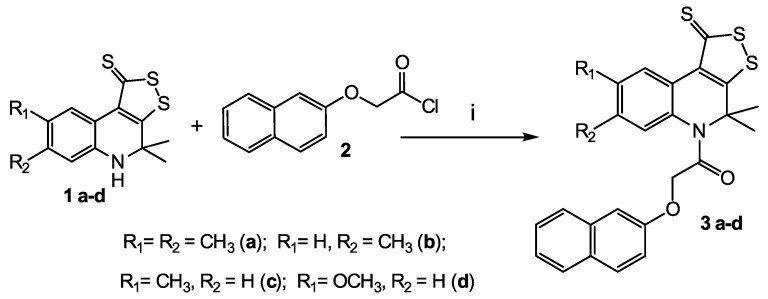
Synthesis of the target compounds (**3 a-d**). Reagents and conditions: (i) dry toluene; dry pyridine, reflux, 5 h.

**Scheme 2 molecules-27-05732-sch002:**
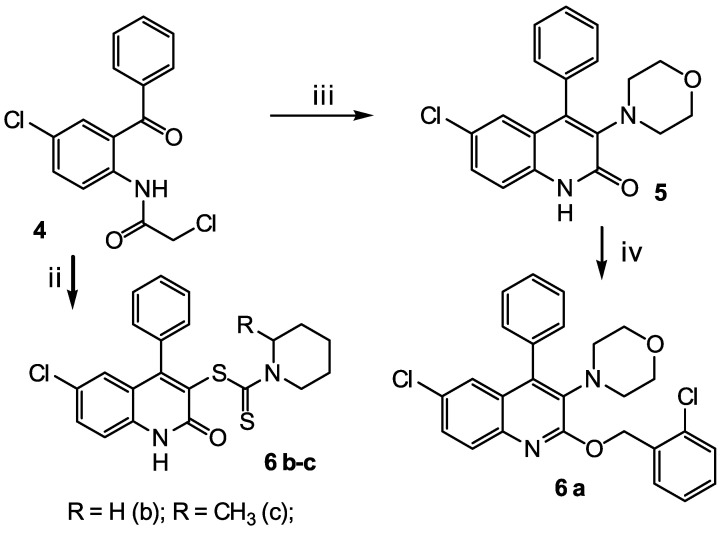
Synthesis of the target compounds (**6 a-c**). Reagents and conditions: (ii) dry DMF, CH${}_{3}$COOK, piperidine-1-yldithiocarboxylic acid (**b**) or 2-methylpiperidine-1-yldithiocarboxylic acid (**c**), 80 °C, 6 h; (iii) dry CH${}_{3}$CN, morpholine, reflux, 3 h; and (iv) dry DMF, CsF, 2-chlorobenzyl chloride, reflux, 4 h.

**Scheme 3 molecules-27-05732-sch003:**
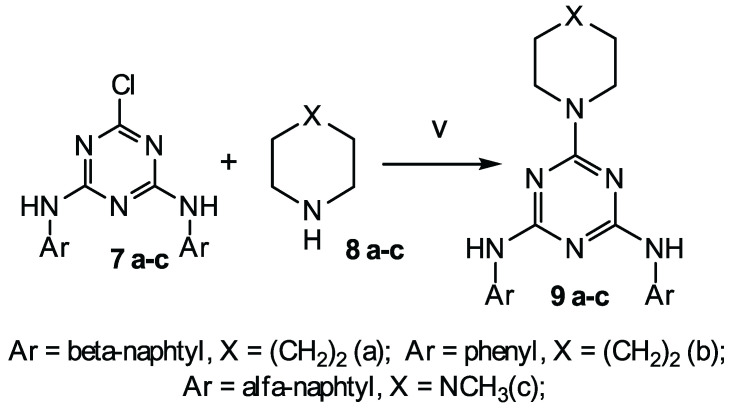
Synthesis of the target compounds (**9 a-c**). Reagents and conditions: (v) CH${}_{3}$CN, K2CO3, 80 °C, 3 h.

**Figure 8 molecules-27-05732-f008:**
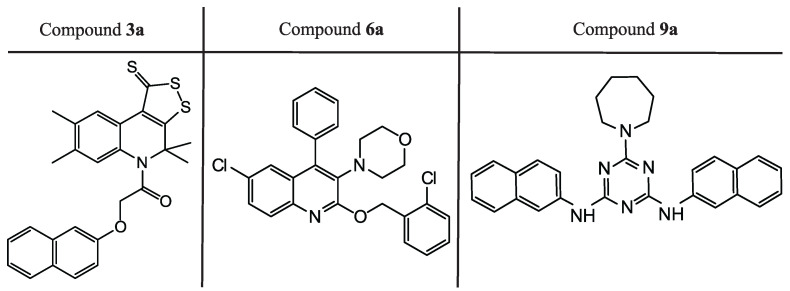
Active compounds with experimentally confirmed activity against SARS-CoV-2 found at the first step of the study.

**Figure 9 molecules-27-05732-f009:**
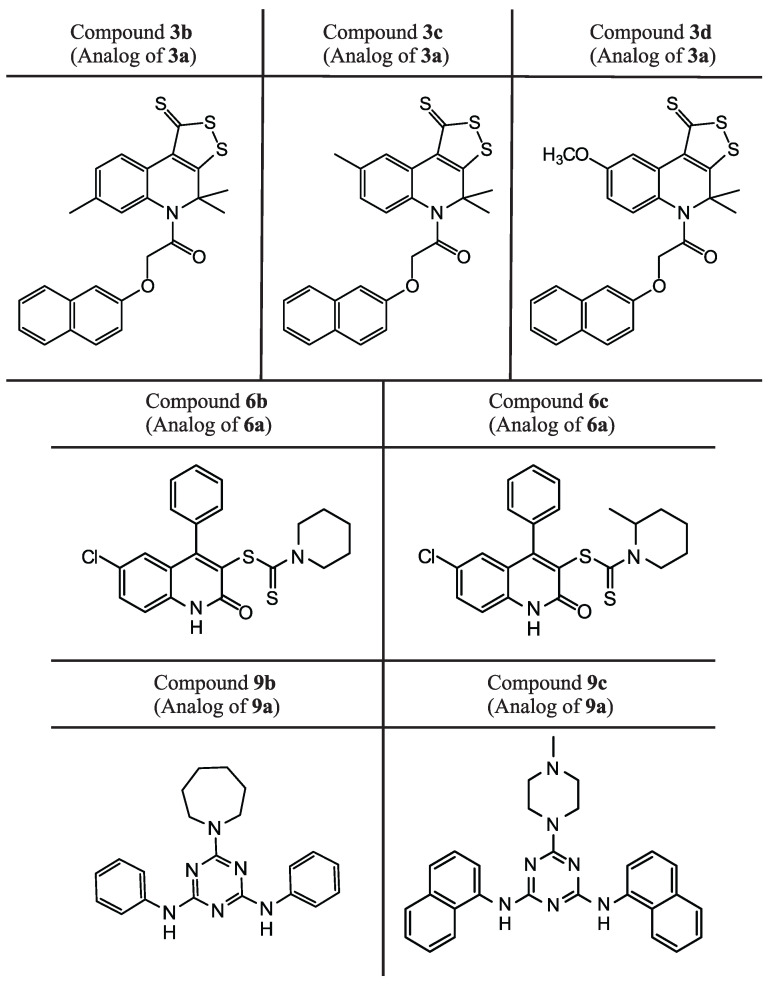
Structures of the ligands presented in Table 2. They are analogs of three compounds presented in Table 1.

**Figure 10 molecules-27-05732-f010:**
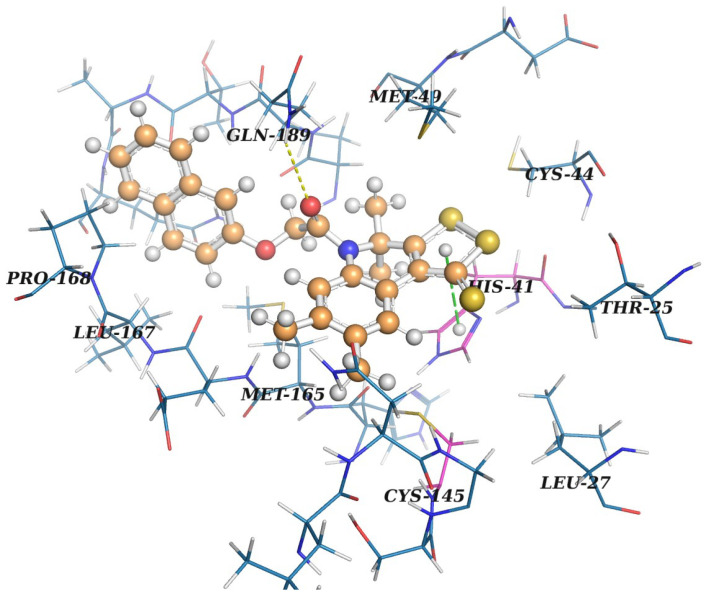
Docking position of compound **3a** in the active site of M${}^{pro}$. The ligand is shown in a “ball-and-stick” model and colored in brown. Protein residues are presented in the “line” model and colored in marine. Residues of a catalytic dyad are colored in magenta. A yellow dashed line indicates an H-bond between the protein and the ligand. Pi-stacking is designated with a green dashed line.

**Table 1 molecules-27-05732-t001:** The calculated and measured characteristics of the leading compounds.

Compound	SOL Score, kcal/mol	$\Delta H_{\mathrm{bind}}$, kcal/mol	EC${}_{50}$, µM	SI
**3a**	−7.54	−33.1	0.51 ± 0.41	>411.5
**6a**	−6.91	−50.8	4.77 ± 1.87	>45.1
**9a**	−6.94	−55.5	18.19 ± 4.20	>11.9
*remdesivir*	-	-	2.94 ± 0.67	56.5

**Table 2 molecules-27-05732-t002:** Chemical analogs of compounds from Table 1 which demonstrated the suppression of SARS-CoV-2 replication in cell culture. Note: **3b-d** are analogs of compound **3a** in Table 1, **6b-c** are analogs of compound **6a**, and **9b-c** are analogs of compound **9a**.

Compound	SOL Score, kcal/mol	$\Delta H_{\mathrm{bind}}$, kcal/mol	EC${}_{50}$, µM	SI
**9b**	−5.76	−53.4	1.04 ± 0.26	>7.57
**3b**	−7.27	−34.5	1.16 ± 0.23	>86.21
**3c**	−7.36	−36.1	2.71 ± 0.91	>36.86
**6b**	−5.57	−28.6	7.62 ± 1.84	>4.75
**3d**	−7.25	−35.0	9.40 ± 1.67	>10.64
**9c**	−6.66	−47.9	19.62 ± 2.94	>1.62
**6c**	−6.03	−45.1	22.40 ± 2.58	>1.2

## Data Availability

Not applicable.

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
