# Peer review of "New Chemicals Suppressing SARS-CoV-2 Replication in Cell Culture"

_molecules, 2022, doi:10.3390/molecules27175732_

Round 1

Reviewer 1 Report

The aim of this study is the computer-aided finding of new inhibitors SARS-CoV-2 MPro inhibitors. The authors applied SOL docking software for virtual screening of the database contained about 19,000 structures of drug-like compounds.

At the first step, three active compounds have been identified; then, the authors performed the search for the structural analogs. All compounds, which were selected on the basis of in silico estimations, were sythesized and their inhibitory activity were tested in Vero E6 cell culture.

As a result, a few compounds with EC50 close or better to the SARS-CoV-2 noncovalent inhibitors published in literature, were discovered.

The manuscript could be accepted for publication after the minor revisions.

1. Scoring function calculated using docking is not the "free energy of binding". It should be replaced by the "estimate of the free energy of binding".  

2. "The higher this energy, the more likely that a given compound will show inhibitory activity in experiments..." The values of the free energy of binding estimates usually are negative; thus, "the higher" is not correct.

3. It is necessary to explain why the 6W63 complex was selected for this study because currently there are dozens SARS-CoV-2 MPro complexes with inhibitors in the PDB.

4. All averages and standard deviations in tables should be rounded to the significant numbers.

5. English language and style are fine but minor grammar check required.

Author Response

Response to Reviewer 1 Comments

Point 1. Scoring function calculated using docking is not the "free energy of binding". It should be replaced by the "estimate of the free energy of binding".  

Response 1: Fixed. The corrected text is highlighted in blue in the new version of the article.

Point 2. "The higher this energy, the more likely that a given compound will show inhibitory activity in experiments..." The values of the free energy of binding estimates usually are negative; thus, "the higher" is not correct.

Response 2: Fixed. Error in translation, corrected to "the lower". The corrected text is highlighted in blue in the new version of the article.

Point 3. It is necessary to explain why the 6W63 complex was selected for this study because currently there are dozens SARS-CoV-2 MPro complexes with inhibitors in the PDB.

Response 3: Fixed. The section 4.1 on protein model preparation has been rewritten. The corrected text is highlighted in blue in the new version of the article.

Point 4. All averages and standard deviations in tables should be rounded to the significant numbers.

Response 4: The calculated values are rounded up to a few decimal places, with which the selected compounds can be distinguished from each other. For the SOL evaluation function it is 2 digits after the decimal point, and for the binding enthalpy it is 1 digit after the decimal point.

Point 5. English language and style are fine but minor grammar check required.

Response 5: Grammar errors have been corrected.

Author Response

Point 1: In Table 1, why authors are not included SOL score, DHbinding for Remdesivir.

Response 1: Remdesivir binds to the RdRp target protein according to structural data obtained by electron microscopy [doi: 10.1126/science.abc1560]. Remdesivir does not bind to Mpro. The additional text is highlighted in blue in the new version of the article.

Point 2: In Scheme 2 and other places, CH3COOK should be given as CH3COOK, CH3CN should be given as CH3CN.

Response 2: Fixed. The corrected text is highlighted in blue in the new version of the article.

Point 3: In lines 296 and 297, Mpro should be given as Mpro.

Response 3: Fixed. The corrected text is highlighted in blue in the new version of the article.